# Antimicrobial Utilization among Neonates and Children: A Multicenter Point Prevalence Study from Leading Children’s Hospitals in Punjab, Pakistan

**DOI:** 10.3390/antibiotics11081056

**Published:** 2022-08-04

**Authors:** Zia Ul Mustafa, Amer Hayat Khan, Muhammad Salman, Syed Azhar Syed Sulaiman, Brian Godman

**Affiliations:** 1Discipline of Clinical Pharmacy, School of Pharmaceutical Sciences, Universiti Sains Malaysia, Gelugor 11800, Penang, Malaysia; 2Department of Pharmacy Services, District Headquarter (DHQ) Hospital, Pakpattan 57400, Pakistan; 3Department of Pharmacy, The University of Lahore, Lahore 54700, Pakistan; 4Strathclyde Institute of Pharmacy and Biomedical Science (SIPBS), University of Strathclyde, Glasgow G4 0RE, UK; 5Department of Public Health Pharmacy and Management, School of Pharmacy, Sefako Makgatho Health Sciences University, Pretoria 0204, South Africa; 6Centre of Medical and Bio-Allied Health Sciences Research, Ajman University, Ajman P.O. Box 346, United Arab Emirates

**Keywords:** point prevalence survey, hospitals, children, antibiotics, Pakistan, AWaRe classification

## Abstract

Antimicrobial resistance (AMR) compromises global health due to the associated morbidity, mortality, and costs. The inappropriate use of antimicrobial agents is a prime driver of AMR. Consequently, it is imperative to gain a greater understanding of current utilization patterns especially in high-risk groups including neonates and children. A point prevalence survey (PPS) was conducted among three tertiary care children’s hospitals in the Punjab province using the World Health Organization (WHO) methodology. Antibiotic use was documented according to the WHO AWaRe classification. Out of a total of 1576 neonates and children, 1506 were prescribed antibiotics on the day of the survey (prevalence = 95.5%), with an average of 1.9 antibiotics per patient. The majority of antibiotics were prescribed in the medical ward (75%), followed by surgical ward (12.8%). Furthermore, 56% of antibiotics were prescribed prophylactically, with most of the antibiotics (92.3%) administered via the parenteral route. The top three indications for antibiotics were respiratory tract infections (34.8%), gastrointestinal infections (15.8%), and prophylaxis for medical problems (14.3%). The three most common antibiotics prescribed were ceftriaxone (25.8%), amikacin (9.2%), and vancomycin (7.9%). Overall, 76.6% of the prescribed antibiotics were from Watch category followed by 21.6% from the Access group. There was a very high prevalence of antibiotic use among hospitalized neonates and children in this study. Urgent measures are needed to engage all the stakeholders to formulate effective ASPs in Pakistan, especially surrounding Watch antibiotics.

## 1. Introduction

Despite widespread advancement in medical science, treatment of infectious diseases is still challenging due to the emergence of antimicrobial resistance (AMR). AMR is not only considered as global health threat but also compromises food security and economic development [1,2,3,4]. More than 4.95 million deaths globally were reported in 2019 in which bacterial AMR played a part, while 1.27 million deaths occurred directly due to AMR [5]. The highest death burden was reported in sub-Saharan Africa and south Asia [5]. AMR is a particular issue in low- and middle-income countries (LMICs); however, the data describing the economic loss as well as mortality due to AMR in these countries are scarce [6,7]. Pakistan is no exception with, for instance, 64% extensively drug-resistant (XDR) cases of tuberculosis were recently identified [8,9,10,11].

Neonates and children are at a comparatively higher risk of bacterial infections due to their fragile physiologic system, greater bacterial exposure, and lack of sufficient immunity [12]. There are concerns with rising rates of AMR in this population in Pakistan [13,14]. Antimicrobials are one of the most frequent classes of medicines prescribed among children and neonates globally [15], with a high percentage of deaths due to AMR globally reported among the newborn [16]. This is driven by the misuse and over-consumption of antimicrobials [2]. Overall, hospitalized settings, particularly among LMICs, are highly vulnerable sites for AMR among neonates and children because of the unnecessary prescribing of antibiotics [17]. This needs to be addressed going forward.

Concerns with the rising rates of AMR and the implications resulted in instigation of the global action plan on AMR by the World Health Organization (WHO). The global action plan stressed the need for awareness, education, and knowledge strengthening through surveillance on AMR as well as optimizing the use of antimicrobial agents [18]. Following this, Pakistan developed its national action plan (NAP) on AMR in 2017 to strengthen all AMR-related activities including national awareness, integrated AMR surveillance, and an estimation of the economic cost of AMR [19]. 

In 2018, a new antimicrobial point prevalence survey (PPS) tool was devised by WHO to estimate the prevalence of antimicrobial utilization especially among LMICs [20]. PPS studies proved a robust and effective methodology to acquire baseline information on antibiotic prescribing among hospitalized patients in specific time frames to guide future quality improvement programs [20,21,22].

Alongside this, the WHO also developed the AWaRe (Access, Watch, and Reserve) classification of antibiotics as a monitoring tool for future quality improvement programs among antimicrobial stewardship activities, especially surrounding Watch and Reserve antibiotics [23,24,25]. The WHO’s target is that at least 60% antibiotics prescribed and utilized are from the Access group [26]. Integration of national AMR surveillance, responsibility, accountability, and monitoring of antibiotics use, as well as the development and implementation of antimicrobial stewardship programs (ASPs), are the key recommendations of situation analysis report on AMR in Pakistan (2018) with collaboration of global antibiotic resistance partnerships (GARP) [27]. However, there are recognized challenges with the implementation of the NAP in Pakistan [28]. Inappropriate prescribing of the antibiotics from prescribers, lack of antibiotics surveillance studies, and the availability of antibiotics without a valid prescription are considered important factors for inappropriate use of antibiotics in Pakistan [27,29,30]. 

Consequently, it is of critical importance to fully assess current antibiotic prescribing practices and administration particularly at hospital level in Pakistan to formulate appropriate ASPs. Whilst PPSs have been conducted predominately among adults in recent times in Pakistan [31,32,33], as well as PPSs and other studies in neonates and children [13,14,17], none of these PPS studies have been undertaken among the tertiary care children’s hospitals in the country. Consequently, the aim of this current study was to conduct a PPS among tertiary care children’s hospitals in the Punjab province, Pakistan, to provide future direction, including establishing ASPs in this critical group where pertinent. This is important since if problems with antimicrobial prescribing are seen in these tertiary hospitals, these would be echoed among the other hospitals in this province and wider throughout Pakistan without similar expertise and back-up.

## 2. Results

### 2.1. Hospital, Ward, Patient, and Antibiotics Prevalence Related Information

A total of 1576 neonates and children were included in the survey, with a bed occupancy rate of 93.2%. The different age groups included neonates (18.9%), infants (30.7%), young children (25.9%), and children (24.3%) as shown in Table 1. In the sample, 95.5% of neonates and children were prescribed antibiotics, highest in Hospital 2 (99.4%). Overall, 2859 antibiotics were prescribed to the study population, with an average of 1.9 antibiotics per patient, highest for Hospital 2 (2.50), with little difference between the age groups (Table 2). The majority of neonates and children (59.8%) were prescribed two antibiotics, while 20.4% were prescribed three or more antibiotics. The majority of antibiotics (92.3%) were prescribed via the parenteral route in the medical ward (75.0%), followed by surgical wards (12.8%) and intensive care units (12.2%). Most of the prescribed antibiotics (56.9%) were for therapeutic purposes while 31.0% were prescribed for prophylaxis. With respect to prophylaxis use, 69.9% were prescribed for medical prophylaxis and 31.1% for surgical prophylaxis. Among patients prescribed antibiotics for surgical prophylaxis, the majority (77.9%) were prescribed for more than one day. Most of the antibiotics (95.2%) prescribed for community acquired infections did not have the reason for their prescription written in the patients’ medical file (92.9%). Moreover, 96.6% antibiotics were prescribed empirically whereas only 3.4% were targeted. In only 30% of occasions were the stop date available in the medical charts. In addition to antibiotics, other antimicrobials prescribed among the surveyed population included antivirals (2.5%), antifungals (2.7%), and antiprotozoals (1.5%).

### 2.2. Prevalence of Prescribed Antimicrobials According to Age Groups

Out of the total 2859 prescribed antibiotics, most were for infants (Table 2). More than three-quarters of total antibiotics (76.6% and 79.6%) prescribed among neonates and infants were in the medical ward for therapeutic purpose (71.9% and 71.8%, respectively). However, there were appreciable differences between the age groups. Among the prophylaxis use of antibiotics, 95.7% were for medical illness among neonates whereas 61.9% of antibiotics were prescribed for surgical prophylaxis among children, reflecting the different patient profiles. Furthermore, 37.4% of neonates’ medical records had the stop date of antibiotics included compared with only 19.9% among the records of young children. 

### 2.3. Antibiotic Resistance Pattern of Identified Bacterial Species 

*Staphylococcus* species including *Staphylococcus aureus* were the most commonly identified bacteria in culture and susceptibility testing, resistant to penicillins and fluoroquinolones (Table 3) but sensitive to vancomycin and linezolid. Other bacteria were *Pseudomonas* species followed by *Klebsiella* species, with the penicillins and 3rd generation cephalosporins resistant to *Pseudomonas* species but susceptible to levofloxacin and cefepime.

### 2.4. Indications for Prescribed Antibiotics 

A little more than one-third (34.8%) of neonates and children were prescribed antibiotics for respiratory tract infections followed by gastrointestinal infections (15.8%) (Table 4). Prophylaxis for medical problems and for surgical procedures were the reason antibiotics were prescribed among 14.3% and 10.2% of children, respectively. Other common illness for antibiotics were sepsis, blood stream, skin, and urinary tract infections (Table 4). 

### 2.5. Indications for Prescribed Antibiotics According to Age Groups

There were differences in the indication for antibiotic prescribing between the different age groups, reflecting the differences in their use (Table 2). Respiratory tract infections were common among infants (40.2%), followed by neonates (26.3%) and young children (19.6%) (Table 5). Gastrointestinal infections were a common indication for antibiotic use in neonates (37.4%) and children (29%). Prophylaxis use of antibiotics for surgical conditions was high in infants (41.5%) compared with neonates (3.2%). Urinary tract infections were a frequent indication among young children (40.3%) compared with neonates (13.4%). 

### 2.6. Details of Prescribed Antibiotics According to ATC Classification

Third generation cephalosporins were the most frequently prescribed antibiotics among the study population, accounting for more than 40% of prescribed antibiotics among neonates and children. Among the third generation cephalosporins, ceftriaxone was the most commonly prescribed antibiotic. Aminoglycosides (amikacin) and glycopeptide antibacterials (vancomycin) were prescribed in 9.2% and 7.9% of the surveyed population, respectively. Other commonly prescribed classes of antibiotics included the macrolides (6.3%), piperacillin plus tazobactam (5.9%), and the aminopenicillins (5.1%) (Table 6).

### 2.7. Detail of Prescribed Antibiotics According to WHO AwaRe Classification

More than three-quarters (76.6%) of prescribed antibiotics belonged to the ‘Watch’ category of antibiotics, with 21.6% from the ‘Access’ group (Table 7), with limited differences between the three hospitals (Figure 1).

## 3. Discussion

We believe the findings will help to establish pediatric ASPs throughout the country. This builds on the findings of Arif et al. (2021) [34] and other recent studies [13,14,17]. Of concern is that almost all hospitalized children and neonates in our study were prescribed antibiotics, similar to the study of Arif et al. [35] where 94.6% of children received at least one antibiotic. In addition, in our previous study among primary and secondary care hospitals in Punjab, almost 97% of neonates and children were prescribed antibiotics [17]. However, this contrasts with lower prevalence rates for antibiotic utilization among neonates and children in other LMICs. This includes South Africa (49.7% of surveyed children), India, (61.5% of surveyed children), and China (66.1%) [35,36,37]. 

Our findings of an average of 1.9 antibiotics prescribed per patient are also similar to our previous study conducted among 16 primary and secondary care hospitals of the Punjab treating neonates and children [17]. However, a recent multi-country study conducted in both high and low-middle income countries had lower rates of antibiotic prescribing per patient in neonates compared to our study [38]. Consequently, this needs to be addressed going forward to reduce future AMR rates in the country.

Another concern is the high use of the parenteral route for administration of antibiotics in our study at more than 90%. This is because the parenteral route can lead to harmful effects including pain at the injection site, phlebitis, and local and systemic infections [39,40,41,42]. In addition, this route can increase the length of stay in hospital, adding to costs [43]. In contrast, a study from Europe reported less than two-thirds of children were prescribed antibiotics via the parenteral route [44]; however, higher rates have been seen in China [45]. 

Our study highlighted that three-quarters of antibiotics were prescribed in the medical unit, followed by surgical units and ICU. These findings are similar to a previous study from India where most of the children admitted in medical units were prescribed antibiotics [37]. In contrast, another study from Turkey documented higher use of antibiotics in pediatric ICUs compared with medical and surgical units [46]. These discrepancies need more careful evaluation as the higher use in medical units may suggest inappropriate prescribing. 

Of equal concern is that antibiotics prescribed for surgical prophylaxis were administered for more than one day in 85% patients. This is a concern as extended prophylaxis increases adverse reactions, costs, and AMR without increasing effectiveness [47]. These findings are similar though to previous studies that highlighted frequent surgical antibiotic prophylaxis for more than one day [17,47,48].

Almost all patients in our study were prescribed antibiotics for community acquired infections. These findings are in contrast to a recent study in Australia where less than half children were prescribed antibiotics for a community acquired infection [49]. Of critical concern in our study was that very few patients were prescribed antibiotics before waiting for culture and susceptibility testing, with the vast majority of children prescribed antibiotics empirically. This is similar to other LMICs [50], though in contrast to a study in the USA where the majority of children were prescribed antibiotics after confirmation from the microbiological testing [51]; and will be followed up in these three hospitals as part of future quality improvement programs.

Another area of concern was that nearly all antibiotics (92.9%) were prescribed without mentioning the reason for their prescription in the medical records. This is similar to the findings by Afriyie et al. in Ghana where the majority of the antibiotics were prescribed without mentioning the reason for this [52]. However, these findings are in contrast to a previous study where approximately half of the hospitalized neonates and children were prescribed antibiotics after mentioning the reason in their medical records [44]. This study though was conducted in Europe as opposed to LMICs. We will be following this up in the future as part of quality improvement programs as such behavior increases inappropriate prescribing of antibiotics.

Respiratory tract infections and gastrointestinal tract infections were common infections among neonates and infants in our study for which antibiotics were prescribed, whereas prophylaxis for surgical and medical problems were common indications among children. This is similar to a previous study from Italy where most of the antibiotics were prescribed for respiratory tract infections followed by prophylactic use for medical patients and surgical procedures [53]. However, contrary to the results of our study, a PPS from Myanmar reported that more than a quarter of children were prescribed antibiotics for surgical prophylaxis [54]. This may reflect the different ages of the children included in the published studies, with a greater use of antibiotics for surgical prophylaxis in children versus neonates. 

*Staphylococcus* species, *Pseudomonas* species, and *Klebsiella* species were common bacterial isolates identified in our survey that were mostly susceptible to vancomycin, levofloxacin, and carbapenems, respectively. These findings are similar to a previous study from South Africa where similar bacterial isolates were reported among hospitalized children indicating similar bacterial flora and antibiotic resistance patterns [36].

Third generation cephalosporins, especially ceftriaxone, were the most frequent class of antibiotics prescribed in our study followed by aminoglycosides. Previous studies among children from Punjab also highlighted frequent use of ceftriaxone among children [17,30,55]. These findings are similar to a previous study from Italy where most of the children prescribed antibiotics were from third generation cephalosporins [53]. Multi-national studies also indicated that ceftriaxone is one of the frequently prescribed antibiotics among children [25]. In contrast to our results, a recent study from Europe reported that most frequent classes of antibiotics prescribed among hospitalized children were cefazolin and amoxicillin-clavulanate [56]. 

Antibiotics included in the ‘Access category’ are key to treat the majority of infections among children without increasing AMR rates [23,24,25,57]; consequently, they should be preferentially prescribed in children. Of concern is that more than three-quarters of the antibiotics prescribed in our study were from the ‘Watch’ category. These findings though are similar to previous study reported by Hsia et al., where the majority of prescribed antibiotics in children were from the ‘Watch’ category [57]. However, they are in contrast to a study reported from Canada where less than half of the antibiotics prescribed among children were from Watch group [58]. In addition, it is also in contrast with a recent study from South Africa where 55.9% of antimicrobials prescribed were from the ‘Access’ group [36]. We will be following this up in future studies to instigate potential ASPs in hospitals treating children to reduce the use of ‘Watch’ antibiotics where antibiotic prescriptions are warranted. 

### Strengths and Limitations

This was the first study conducted among all three tertiary care children’s hospital of the province Punjab to extract baseline information on antimicrobial use among inpatients. We believe that our findings will provide comprehensive information to the policy makers and healthcare providers to formulate institutional treatment guidelines, upgrade diagnostic facilities especially with the recent pandemic, and implement pediatric ASPs. 

We are aware of a number of limitations with our study. Firstly, we collected data from only three tertiary children’s hospitals in the Punjab and did not collect data from other provinces. Consequently, we are unable to fully generalize the finding of this study. In addition, the inherent problems with PPS studies are that all the relevant data may not be recorded in patient’s notes. However, despite this, we believe our findings are robust providing future direction.

## 4. Materials and Methods

### 4.1. Study Design 

A multicenter PPS was undertaken among Tertiary Care children’s hospitals in the Punjab Province according to the WHO standardized methodology, which was built on the Global PPS and the European Centre for Disease Prevention and Control (ECDC) PPS studies [20,21,59]. As continuous antibiotics use surveillance is more time- and resource-consuming, regular PPSs are seen as easier to execute. According to the PPS methodology, data are collected at the hospital, ward, and patient level. 

### 4.2. Study Time and Settings

We performed this initial study in the largest province in Pakistan, Punjab, where the public sector health department, government of the Punjab, is divided into tertiary care/teaching hospitals named as Specialized Healthcare and Medical Education Department (SHCME) and Primary and Secondary Healthcare Department (P&SHD). Under the administrative control of SHCME, 49 tertiary/teaching hospitals are currently serving patients throughout the province. From these 49 tertiary care hospitals, only three tertiary care hospitals are specified for the neonates and children. Consequently, we initially conducted our study in these three tertiary cares children’s hospitals between 10 December 2021 and 5 January 2022 to provide a baseline to monitor prescribing in less specialized centers. As stated, if problems with antimicrobial prescribing are encountered in these three tertiary hospitals, these would be echoed among the other hospitals in the province. These three tertiary hospitals have been designated anonymously as H1, H2, and H3 in line with other PPS studies involving multiple hospitals [17,20]. These children’s hospitals were equipped with all the necessary facilities to provide tertiary level care. All the hospitals had different wards to take care of different pediatric populations, including neonatal medical wards, neonatal intensive care units (NICU), pediatric medical wards, pediatric surgical wards, and pediatric intensive care wards. Furthermore, all the hospitals have trained healthcare professionals as well as laboratory facilities and medicines provided free-of-charge to patient’s families to optimize healthcare. Different age groups of the study population were neonates (1–28 days), infants (29 days–1 year), young children (>1–5 years) and children (>5–12 years) as per previous PPS studies [17].

### 4.3. Data Collection Procedure

Healthcare professionals from the children’s hospitals were invited by the team of investigators to participate in this study. The principal investigator (Z.U.M.) briefed the team about the purpose of the study, its methodology, as well as inclusion and exclusion criteria of the children before initiation of the study. Data collection forms were provided to team members to collect the necessary information. The data collection team visited different wards of the participant’s hospitals on the day of survey at 8:00 a.m. and asked the clinical staff about the number of admitted patients at that time. Subsequently, the medical records of the pertinent inpatients were searched thoroughly by the investigators to obtain the required data. In case of any inquiry, the clinical staff were contacted during data collection. The data collection form was divided into three sections


**Section I**


This section collected information relating to the hospital including its name, ownership, administrative control, number of total functional beds, various facilities, and operations in the hospital as well as the availability of antibiotics in the hospital.


**Section II**


This section collected information related to various wards in the hospital. This included the functionality of the wards including their subspeciality and the total number of beds in each ward.


**Section III**


In this section, patient-related information was gathered. This included the age, gender, reason of hospitalization, diagnosis, and details of any surgical procedure of the admitted neonates and children. Information concerning the indications including therapeutic or prophylaxis use of the antibiotics was also collected, as well as, if prophylaxis, whether this was medical or surgical prophylaxis. For surgical prophylaxis, the duration of prescribed antibiotics was also recorded. This section also collected detail information about the antibiotics prescribed by their INN (International Nonproprietary Name) and ATC code [60], their route of administration, the rationale of the antibiotics being prescribed, as well as stop date/time. Subsequently antibiotics use was broken down by the WHO’s AWaRe classification.

### 4.4. Inclusion and Exclusion Criteria

All inpatient children and neonates who had stayed overnight in the respective wards at 8:00 a.m. on the day of survey and prescribed with any antibiotic for bacterial-infection through oral, parenteral, or rectal were included in current study. All children that had visited the emergency departments or stayed in hospital for short duration, including patients on dialysis or patients on long-term medication for chronic illness, were excluded from current study. Moreover, antibiotics prescribed after 8:00 a.m. on the day of survey or via the topical route were excluded from the study. 

### 4.5. Statistical Analysis

All the data were entered on SPPS version 22. Continuous data were expressed as means while categorical data were expressed as frequencies and percentages.

## 5. Conclusions

There was a very high prevalence of antimicrobial use among hospitalized neonates and children in our study. In addition, antibiotics were typically prescribed empirically via the parenteral route without confirmation from culture and susceptibility testing. This is a concern as this can extend hospital stay adding to the costs alongside increasing AMR. Extended prophylactic use of antibiotics in medical illness and surgical procedures is also a concern requiring urgent activities as part of any quality improvement programs. Other urgent targets are the lack of stop date/times in patient’s notes. Finally, there are high rates of prescribing of antibiotics from the WHO ‘Watch’ category. All of these areas need to be prioritized in future quality improvement programs, and we will be monitoring this starting in the three tertiary hospitals surveyed.

## Figures and Tables

**Figure 1 antibiotics-11-01056-f001:**
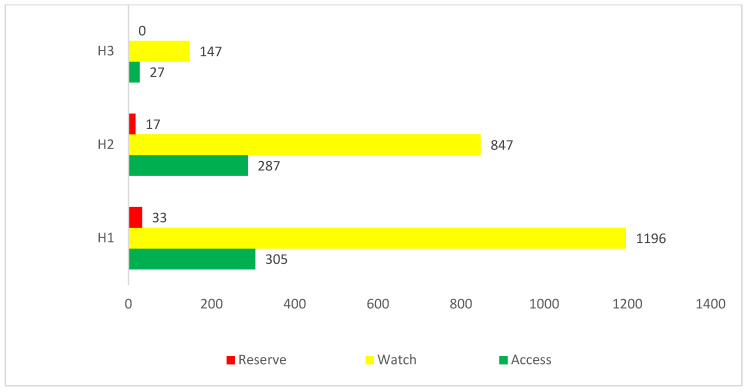
Antibiotics use according to WHO AWaRe classification.

**Table 1 antibiotics-11-01056-t001:** Hospital, ward, patient, and antibiotics prevalence related information.

Variables N (%)	H1	H2	H3	Total N (%)
Total bed in hospital	1050	570	250	1870
Total bed in children ward	1010	510	170	1690
Total patients in children ward at 8:00 AM	987	464	125	1576 (93.2)
Total No. of patients prescribed antibiotics and %	936 (94.8)	461 (99.4%)	109 (87.2%)	1506 (95.5)
**Age (Number and %)**				
Neonates (0–28 days)	136 (14.5)	142 (30.8)	8 (7.3)	286 (19.0)
Infants (29 days–1 year)	337 (36.0)	83 (18.0)	42 (38.5)	462 (30.7)
Young child (˃1–5 years)	252 (26.9)	113 (24.5)	26 (23.8)	391 (25.9)
Child (˃5–12 years)	211 (22.6)	123 (26.6)	33 (30.3)	367 (24.3)
Total no. of prescribed antibiotics and per patient	1534 (1.64)	1151 (2.50)	174 (1.60)	2859 (1.90)
**No. (and %) of Antibiotics per** **patient**				
One antibiotic	354 (23.0)	156 (13.5)	51 (29.3)	561 (19.6)
Two antibiotics	1032 (67.3)	572 (49.7)	108 (62.0)	1712 (59.9)
Three antibiotics or above	148 (9.7)	423 (36.8)	15 (8.7)	586 (20.4)
**Other anti-infective agents**				
Antiviral	52 (32.0)	23 (53.5)	03 (60.0)	78 (2.5)
Antifungal	79 (48.8)	04 (9.3)	01 (20.0)	84 (2.7)
Antiprotozoal	31 (19.2)	16 (37.2)	01 (20.0)	48 (1.5)
**Gender**				
Male	526 (56.2)	174 (37.8)	64 (58.7)	764 (50.7)
Female	410 (43.8)	287 (62.2)	45 (41.3)	742 (49.3)
**Route of administration**				
Oral	164 (10.7)	45 (4.0)	13 (7.5)	222 (7.7)
Parenteral	1370 (89.3)	1106 (96.0)	161 ((92.5)	2637 (92.3)
**Sub-specialty**				
Medical	1267 (82.6)	752 (65.3)	126 (72.4)	2145 (75.0)
Surgical	175 (11.4)	156 (13.6)	36 (20.6)	367 (12.8)
ICU	92 (5.9)	243 (21.1)	12 (6.9)	347 (12.2)
**Indications**				
Therapeutic use	962 (62.7)	571 (49.6)	94 (54.0)	1627 (56.9)
Prophylaxis use	356 (23.2)	468 (40.6)	63 (36.2)	887 (31.0)
Unknown	216 (14.1)	112 (9.8)	17 (9.8)	345 12.1)
**Indications for prophylaxis**				
Medical	257 (72.2)	327 (69.9)	36 (57.1)	620 (69.9)
Surgical	99 (27.8)	141 (30.1)	27 (42.9)	267 (31.1)
**Surgical prophylaxis**				
Single dose	31 (31.3)	0	0	31 (11.6)
One day	17 (17.2)	07 (4.9)	04 (14.8)	28 (10.4)
More than one day	51 (51.5)	134 (95.0)	23 (85.2)	208 (77.9)
**Indication of infection**				
Community acquired	1457 (94.9)	1113 (96.7)	152 (87.3)	2722 (95.2)
Hospital acquired	77 (5.0)	38 (3.3)	22 (12.7)	137 (4.8)
**Reasons on notes**				
No	1387 (90.4)	1097 (95.3)	171 (98.2)	2655 (92.9)
Yes	147 (9.6)	54 (4.7)	03 (1.7)	204 (7.1)
**Stop date**				
Yes	573 (37.3)	237 (20.6)	41 (23.5)	851 (29.7)
No	961 (62.7)	914 (79.4)	133 (76.4)	2008 (70.3)
**Types of therapy**				
Empirical therapy	1480 (96.5)	1118 (97.1)	165 (94.8)	2763 (96.6)
Targeted therapy	54 (3.5)	33 (2.9)	09 (5.1)	96 (3.4)

**Table 2 antibiotics-11-01056-t002:** Details of prescribed antibiotics according to age groups.

Variables N (%)	Neonates	Infants	Young Child	Child
**No. of Antibiotics and per patient prescribed antibiotics**	569 (1.99)	852 (1.84)	771 (1.97)	667 (1.82)
**Sub-specialty (number and %)**				
Medical	436 (76.6)	678 (79.6)	565 (73.3)	466 (69.8)
Surgical	5 (0.9)	68 (8.0)	136 (17.6)	158 (23.7)
ICU	128 (22.5)	106 (12.4)	70(9.0)	43 (6.4)
**Indications for use**				
Therapeutic use	409 (71.9)	612 (71.8)	406 (52.6)	200 (30.0)
Prophylaxis use	118 (20.7)	178 (20.9)	281 (36.4)	310 (46.5)
Unknown	42 (7.4)	62 (7.2)	84 (10.9)	157 (23.5)
**Indications for prophylaxis**				
Medical	113 (95.7)	130 (73.0)	185 (65.9)	192 (61.9)
Surgical	5 (4.2)	48 (27.0)	96 (34.1)	118 (38.0)
**Surgical prophylaxis**				
Single dose	1 (20.0)	7 (14.6)	14 (14.6)	09 (7.6)
One day	1 (20.0)	08 (16.7)	16 (16.6)	03 (2.5)
More one day	3 (60.0)	33 (3368.7)	66 (68.8)	106 (89.8)
**Indication of infection**				
Community acquired	560 (98.4)	826 (97.0)	734 (95.2)	602 (90.2)
Hospital acquired	09 (1.6)	26 (3.0)	37 (4.8)	65 (9.8)
**Reasons on notes**				
No	508 (89.3)	824 (96.7)	718 (93.1)	605 (90.7)
Yes	61 (10.7)	28 (3.3)	53 (6.9)	62 (9.3)
**Stop date**				
Yes	213 (37.4)	191 (22.4)	153 (19.9)	294 (44.0)
No	356 (62.6)	661 (77.6)	618 (80.1)	373 (56.0)
**Types of therapy**				
Empirical therapy	556 (97.7)	811 (95.1)	744 (96.5)	652 (97.7)
Targeted therapy	13 (2.3)	41 (4.8)	27 (3.5)	15 (2.2)

**Table 3 antibiotics-11-01056-t003:** Antibiotic resistance and antibiotic susceptibility profiles of commonly identified bacterial species.

Common Identified Bacteria	Common Resistant Antibiotics	Common Sensitive Antibiotics	H1	H2	H3	Total N (%)
*Staphylococcus* species	Ampicillin, amoxicillin, ciprofloxacin, levofloxacin,	Vancomycin, linezolid	24	05	-	29 (30.2)
*Pseudomonas* species	Penicillins, 3rd generation cephalosporins, amikacin	Levofloxacin, cefepime	11	10	-	21 (21.9)
*Klebsiella* species	3rd generation cephalosporins such as cefotaxime, ceftriaxone	Cephoperazone + beta-lactamase inhibitor, meropenem, imipenem	04	11	05	20 (20.9)
*Escherichia coli*	Penicillin, amoxicillin	Carbapenems such as meropenem, imipenem	11	-	04	15 (15.7)
*Shigella* species	Ampicillin, amoxicillin	Ceftriaxone, ciprofloxacin		05		05 (5.2)
Others	-	-	04	02	-	06 (6.2)

**Table 4 antibiotics-11-01056-t004:** Indications for prescribed antibiotics among the study participants.

Infection Type	H1 (%)	H2 (%)	H3 (%)	Total N (%)
Respiratory tract infections	346 (66.0)	147 (28.0)	31 (5.9)	524 (34.8)
Gastrointestinal infections	152 (63.9)	69 (28.9)	17 (7.1)	238 (15.8)
Prophylaxis for medical problems	116 (53.7)	87 (40.2)	13 (6.0)	216 (14.3)
Prophylaxis for surgical diseases	96 (62.3)	42 (27.3)	16 (10.4)	154 (10.2)
Blood stream infection	48 (51.6)	36 (38.7)	09 (9.6)	93 (6.1)
Skin and soft tissue infections	57 (63.3)	27 (30.0)	06 (6.6)	90 (6.0)
Sepsis	46 (54.1)	23 (27.0)	16 (18.8)	85 (5.6)
Urinary tract infections	48 (71.6)	19 (28.3)	0	67 (4.4)
Others	27 (69.2)	11 (28.2)	01 (2.5)	39 (2.6)

NB: Medical problems indicated dyspepsia, anemia, cystic fibrosis, and autoimmune diseases.

**Table 5 antibiotics-11-01056-t005:** Indications for prescribed antibiotics according to age groups.

Variables N (%)	Neonates (%)	Infants (%)	Young Child (%)	Child (%)
Respiratory tract infections	138 (26.3)	211 (40.2)	103 (19.6)	72 (13.7)
Gastrointestinal infections	89 (37.4)	52 (21.8)	28 (11.7)	69 (29.0)
Prophylaxis for medical problems	41 (19.0)	83 (38.4)	47 (21.7)	45 (20.8)
Prophylaxis for surgical diseases	05 (3.2)	64 (41.5)	37 (24.0)	48 (31.1)
Blood stream infection	12 (12.9)	11 (11.8)	42 (45.1)	31 (33.3)
Skin and soft tissue infections	09 (10.0)	23 (25.5)	18 (20.0)	40 (44.4)
Sepsis	37 (43.5)	14 (60.8)	23 (27.0)	11 (12.9)
Urinary tract infections	09 (13.4)	14 (20.9)	27 (40.3)	17 (25.4)
Others	13 (33.3)	08 (20.5)	06 (15.4)	12 (30.7)

NB: %s in each column equate to the %s for each indication by age group.

**Table 6 antibiotics-11-01056-t006:** Detail of prescribed antibiotics according to ATC classification.

ATC class	Name of Antibiotics (ATC code)	H1	H2	H3	Total (%)
Third-generation cephalosporin	Ceftriaxone (J01DD04)	419	277	44	740 (25.8)
	Cefotaxime (J01DD01)	119	83	28	230 (8.0)
	Ceftazidime (J01DD02)	23	31	13	67 (2.3)
	Cefixime (J01DD08)	27	-	-	27 (0.9)
	Cephoperazone + beta-lactamase inhibitor (salbactam) (J01DD12)	20	73	14	107 (3.7)
Aminoglycoside	Amikacin (D06AX12)	13	112	21	264 (9.2)
Glycopeptide antibacterials	Vancomycin (J01XA01)	177	46	05	228 (7.9)
Macrolides	Azithromycin (J01FA10)	53	69	03	125(4.3)
	Clarithromycin (J01FA09)	24	27	07	58 (2.0)
Piperacillin and enzyme inhibitor	Piperacillin + enzyme inhibitor (tazobactam) (J01CR05)	89	71	11	171 (5.9)
Aminopenicillins	Ampicillin (J01CA01)	64	82	02	148 (5.1)
Amoxicillin and beta-lactamase inhibitor	Amoxicillin + beta-lactamase inhibitors (clavulanate) (J01CR02)	74	56	-	130 (4.5)
Carbapenems	Meropenem (J01DH02)	49	57	12	118 (4.1)
	Imipenem and cilastatin (J01DH51)	09	04	-	13 (0.4)
Fluoroquinolones	Ciprofloxacin (J01MA02)	63	43	04	110 (3.8)
	Moxifloxacin (J01MA14)	14	03	-	17 (0.5)
Imidazole derivatives	Metronidazole (J01XD01)	58	26	02	86 (3.0)
Fourth-generation cephalosporins	Cefepime (J01DE01)	53	30	04	87 (3.0)
Penicillins with extended spectrum	Amoxicillin (J01CA04)	13	37	04	54 (1.8)
Other antibacterials	Linezolid (J01XX08)	33	17	-	50 (1.7)
First generation cephalosporins	Cefradine (J01DB09)	11	-	-	11 (0.3)
Others	-	11	07	-	18 (0.6)

**Table 7 antibiotics-11-01056-t007:** Detail of prescribed antibiotics according two WHO AwaRe classification.

AwaRe Category	H1	H2	H3	Total N (%)
Access	305	287	27	619 (21.7)
Watch	1196	847	147	2190 (76.6)
Reserve	33	17	0	50 (1.7)

## Data Availability

Available on reasonable request from the corresponding authors.

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
