# Peer review of "Antimicrobial Utilization among Neonates and Children: A Multicenter Point Prevalence Study from Leading Children’s Hospitals in Punjab, Pakistan"

_antibiotics, 2022, doi:10.3390/antibiotics11081056_

Round 1
Reviewer 1 Report
A simple study, well presented.
I am concerned about the antibiotic resistances reported eg for Pseudomonas, penicillins and 3rd gen cephalosporins. This is not of interest as these organisms are intrinsically resistant to these antibiotics and so there is no need to comment. Likewise, penicillin for E. coli. Please discuss with a microbiologist and only comment on relevant antibiotic problems.
Please include S. aureus as their own category as this is the pathogen group. The coagulase negative Staphs may not be as important.
Author Response
Open Review
( ) I would not like to sign my review report
(x) I would like to sign my review report
English language and style
( ) Extensive editing of English language and style required
(x) Moderate English changes required
( ) English language and style are fine/minor spell check required
( ) I don't feel qualified to judge about the English language and style
|
Yes |
Can be improved |
Must be improved |
Not applicable |
|
|
Does the introduction provide sufficient background and include all relevant references? |
(x) |
( ) |
( ) |
( ) |
|
Are all the cited references relevant to the research? |
(x) |
( ) |
( ) |
( ) |
|
Is the research design appropriate? |
(x) |
( ) |
( ) |
( ) |
|
Are the methods adequately described? |
(x) |
( ) |
( ) |
( ) |
|
Are the results clearly presented? |
(x) |
( ) |
( ) |
( ) |
|
Are the conclusions supported by the results? |
(x) |
( ) |
( ) |
( ) |
Comments and Suggestions for Authors
- A) A simple study, well presented.
Author reply – Thank you for these kind comments – appreciated
- B) Moderate English changes required
Thank you – now been through the manuscript and updated where we can based on the experience of one of the co-authors who is a native English speaker with more than 450 publications in peer-reviewed journals since 2008. We hope this is now OK.
- C) I am concerned about the antibiotic resistances reported eg for Pseudomonas, penicillins and 3rd gen cephalosporins. This is not of interest as these organisms are intrinsically resistant to these antibiotics and so there is no need to comment. Likewise, penicillin for E. coli. Please discuss with a microbiologist and only comment on relevant antibiotic problems.
Author reply
Thank you for the comment. We are agreed with your valuable comment but in this study, we have only reported what we observed during the survey. The local antibiotic resistant pattern is very limitedly reported from these health settings. Consequently, we are hopeful that these findings would be of assistance to key stakeholder groups in these hospitals and wider in order to better understand local antibiotic resistance patterns and their implications. We hope this is acceptable.
- D) Please include S. aureus as their own category as this is the pathogen group. The coagulase negative Staphs may not be as important.
Author reply – Thank you for this comment. We would like to keep the Staphylococcus group together as this helps to compare and contrast with the other key groups as undertaken in Table 3. We have though indicated that this group contains S aureus. We hope this is OK.
Reviewer 2 Report
Major Comments
In introduction, line 95 please expand on study purpose. “Provide future direction” is too vague.
Table 4 please define what constitutes a “medical problem” as a foot note and modify to underlying condition or other more descriptive term
In tables often the percentages do not add up to 100% please modify as needed
In the Tables the S. No. column is not necessary, recommend removing
Table 6: Please specify what enzyme inhibitor is paired with piperacillin, correct for other combo agents as well throughout tables and manuscripts
In Tables I do not think the ATC code is necessary, recommend removing
Minor Comments
Line 137 “spec” should be changed to species or spp. and should not be in italics
Line 137 sensitivity testing should be changed to susceptibility testing, please correct throughout manuscript where relevant i.e. sensitive should be susceptible
Line 138 Pseudomonas should be in italics
Line 140 species should NOT be in italics, correct throughout the manuscript and in tables
Author Response
Open Review
( ) I would not like to sign my review report
(x) I would like to sign my review report
English language and style
( ) Extensive editing of English language and style required
(x) Moderate English changes required
( ) English language and style are fine/minor spell check required
( ) I don't feel qualified to judge about the English language and style
|
Yes |
Can be improved |
Must be improved |
Not applicable |
|
|
Does the introduction provide sufficient background and include all relevant references? |
( ) |
(x) |
( ) |
( ) |
|
Are all the cited references relevant to the research? |
(x) |
( ) |
( ) |
( ) |
|
Is the research design appropriate? |
( ) |
(x) |
( ) |
( ) |
|
Are the methods adequately described? |
( ) |
(x) |
( ) |
( ) |
|
Are the results clearly presented? |
( ) |
(x) |
( ) |
( ) |
|
Are the conclusions supported by the results? |
(x) |
( ) |
( ) |
( ) |
Comments and Suggestions for Authors
Major Comments
- A) In introduction, line 95 please expand on study purpose. “Provide future direction” is too vague.
Author reply:
Thank you for the comment – now updated
- B) Table 4 please define what constitutes a “medical problem” as a foot note and modify to underlying condition or other more descriptive term
Author reply:
Thank you for the comment. We have added foot note of common medical problems for which prophylactic use of antibiotics were observed. We hope this is now OK
- C) In tables often the percentages do not add up to 100% please modify as needed.
Author reply:
Thank you for the comment. We have updated tables with the correct percentages.
- C) In the Tables the S. No. column is not necessary, recommend removing
Author reply:
Thank you for the comment. We have removed serial number from the tables.
D)
Table 6: Please specify what enzyme inhibitor is paired with piperacillin, correct for other combo agents as well throughout tables and manuscripts
Author reply:
Thank you. We have update manuscript as per your comment.
- E) In Tables I do not think the ATC code is necessary, recommend removing
Author reply:
Thank you. We have seen in many PPS studies that ATC code was mentioned along with generic name of the antibiotics. We think this is important as the ATC code is universal specifically identifying each antibiotic. Consequently, if agreeable we prefer to keep this column
Minor Comments
- A) Line 137 “spec” should be changed to species or spp. and should not be in italics
Author reply:
Thank you. We have updated it.
- B) Line 137 sensitivity testing should be changed to susceptibility testing, please correct throughout manuscript where relevant i.e. sensitive should be susceptible.
Author reply:
Thank you. We have updated manuscript as per your comment.
- C) Line 138 Pseudomonas should be in italics
Author reply:
Thank you. We have updated it.
- D) Line 140 species should NOT be in italics, correct throughout the manuscript and in tables
Author reply:
Thank you. We have updated it.